# Histological Assessment of Endochondral Ossification and Bone Mineralization

**Tomoka Hasegawa** [1], **Hiromi Hongo** [1], **Tomomaya Yamamoto** [1,2], **Takafumi Muneyama** [1], **Yukina Miyamoto** [1] **and Norio Amizuka** [1,*]

1. Developmental Biology of Hard Tissue, Faculty of Medicine, Hokkaido University, Sapporo 060-8586, Japan
2. Northern Army Medical Unit, Camp Makomanai, Japan Ground Self-Defense Forces, Sapporo 005-8543, Japan
* Correspondence: amizuka@den.hokudai.ac.jp; Tel.: +81-11-706-4223

**Abstract:** Finely tuned cartilage mineralization, endochondral ossification, and normal bone formation are necessary for normal bone growth. Hypertrophic chondrocytes in the epiphyseal cartilage secrete matrix vesicles, which are small extracellular vesicles initiating mineralization, into the intercolumnar septa but not the transverse partitions of the cartilage columns. Bone-specific blood vessels invade the unmineralized transverse septum, exposing the mineralized cartilage cores. Many osteoblast precursors migrate to the cartilage cores, where they synthesize abundant bone matrices, and mineralize them in a process of matrix vesicle-mediated bone mineralization. Matrix vesicle-mediated mineralization concentrates calcium (Ca) and inorganic phosphates (Pi), which are converted into hydroxyapatite crystals. These crystals grow radially and are eventually get out of the vesicles to form spherical mineralized nodules, leading to collagen mineralization. The influx of Ca and Pi into the matrix vesicle is regulated by several enzymes and transporters such as TNAP, ENPP1, PiT1, PHOSPHO1, annexins, and others. Such matrix vesicle-mediated mineralization is regulated by osteoblastic activities, synchronizing the synthesis of organic bone material. However, osteocytes reportedly regulate peripheral mineralization, e.g., osteocytic osteolysis. The interplay between cartilage mineralization and vascular invasion during endochondral ossification, as well as that of osteoblasts and osteocytes for normal mineralization, appears to be crucial for normal bone growth.

**Keywords:** matrix vesicle; mineralization; bone; endochondral ossification; osteoblast





## 1. Introduction

The growth of long bone depends on endochondral ossification, which can be sequentially divided into cartilage matrix mineralization, vascular invasion into the epiphyseal cartilage to expose the mineralized cartilage matrix, osteoblastic migration into the mineralized cartilage cores, and bone deposition to form the primary trabeculae. Hypertrophic chondrocytes play a key role in normal cartilage mineralization, and subsequently in endochondral ossification. These hypertrophic chondrocytes secrete matrix vesicles, extracellular small vesicles that initiate mineralization, and also produce vascular endothelial growth factor (VEGF) allowing the vascular endothelial cells to invade the epiphyseal cartilage. Cartilage mineralization is involved in the modeling of long bones and their changes of shape and size, i.e., the development and growth of the metaphyseal trabeculae. Finely tuned interplays among chondrocytes, vascular endothelial cells, osteoclasts (chondroclasts), and osteoblasts is apparently necessary for adequate endochondral ossification [1].

Bone is a mineralized tissue composed of calcium phosphates and organic materials such as collagen and proteoglycans. There are two phases of bone mineralization: primary and secondary. Primary mineralization is achieved by osteoblasts. Osteoblasts also produce a large amount of matrix vesicles, which mineralize in nodules (globular assemblies of hydroxyapatite crystals) and then extend into the collagen fibrils secreted by the osteoblasts. In contrast to primary mineralization, secondary mineralization is the process whereby the

mineral density of bone increases after primary mineralization. It is postulated that secondary mineralization is regulated through physical crystal maturation, and by the cellular activities of osteocytes embedded in the bone matrix. However, the exact mechanism of secondary mineralization is not yet fully understood.

Histological processes of primary mineralization in the bones can be divided into two phases: matrix vesicle-mediated mineralization and collagen mineralization. In matrix vesicle-mediated mineralization, osteoblasts appear to regulate the secretion speed and the amount of matrix vesicle according to the synthesis of bone matrix. The discovery of matrix vesicles was a breakthrough in the field of bone mineralization [2–8], and many membrane transporters and enzymes related to matrix vesicle-mediated mineralization have recently been discovered. In addition to matrix vesicle-mediated mineralization, recent reports have suggested that osteocytes putatively regulate the mineralization in the periphery. As osteoblasts and osteocytes are directly connected to each other by means of their cytoplasmic processes, bone mineralization may be regulated by the interplay of osteoblasts and osteocytes. Updated knowledge of the matrix vesicles and osteocytic network in bone mineralization may deepen the understanding of mineral metabolism in bones.

In this review, we present the ultrastructural and histological aspects of endochondral ossification, matrix vesicle-mediated mineralization, and osteocytic regulation of bone mineralization.

## 2. Histological Aspects on Endochondral Ossification

### 2.1. Cartilage Mineralization by Hypertrophic Chondrocytes

Epiphyseal cartilage can be divided into three distinctive zones: resting, proliferating, and hypertrophic zones. Chondrocytes form the longitudinal columns in the proliferative and hypertrophic zones, but the proliferative chondrocytes synchronously enlarge in the hypertrophic phenotype [1]. Parathyroid hormone (PTH)-related peptide (PTHrP) has been reported to regulate hypertrophic differentiation of chondrocytes by mediating the Indian hedgehog (IHH)/PTHrP negative feedback [9]. IHH expressed in the prehypertrophic zone (the upper region of the hypertrophic zone) stimulates PTHrP expression in the early differentiation stage of chondrocytes. PTHrP promotes the proliferation activity of chondrocytes by binding to the common receptor of PTH and PTHrP (PTH/PTHrP receptor) in the proliferative zone. PTHrP alternatively inhibits the hypertrophic phenotype of chondrocytes, and IHH expression is then turned off. In addition to IHH/PTHrP negative feedback, another important regulatory factor in chondrocyte proliferation is fibroblast growth factor receptor 3 (FGFR3). Point mutations in FGFR3 cause achondroplasia and thanatophoric dysplasia by continuous activation of the transcription factor Stat1 [10,11]. FGFR3 signaling has also been proposed to increase the pool of proliferating cells by stimulating chondrocytes in the resting zone and promoting their transit to the proliferative zone [12,13]. Thus, the action of IHH/PTHrP and FGFR3 may be essential for chondrocyte proliferation and differentiation [14]. Hypertrophic chondrocytes have large and translucent cell bodies and produce type I and X collagens, tissue nonspecific alkaline phosphatase (TNAP), proteoglycan, and osteopontin [15–19]. Hypertrophic chondrocytes do not proliferate but acquire mineralization ability in the cartilage matrix. Hypertrophic chondrocytes also reportedly secrete VEGF, an angiogenic molecule that has been implicated in matrix metabolism enabling vascular invasion of the epiphyseal cartilage [20]. Hypertrophic chondrocytes of the epiphyseal cartilage secrete matrix vesicles, in which crystalline calcium phosphates appear, forming hydroxyapatite crystals that grow and eventually break through the membrane to form mineralized nodules in the cartilage matrix. Hypertrophic chondrocytes deposit matrix vesicles in the intercolumnar septae but not in the transverse partitions, consequently forming mineralized longitudinal septae and unmineralized transverse partitions. The regular distribution of mineralized cartilage matrix in the longitudinal intercolumnar septum allows the vertical invasion of vascular endothelial cells, which infiltrate into the cartilage by penetrating the unmineralized transverse partitions. After the formation of these calcified

cartilage cores exposed to bone tissues, many osteoblast precursors migrate and attach to the mineralized cartilage cores to deposit abundant organic bone matrices including type I collagen, osteocalcin, osteopontin, and so forth, thereby forming the primary trabeculae. Thus, the process of endochondral ossification involves a well-defined series of events which include the invasion of vascular endothelial cells, osteogenic cell migration, new bone deposition onto the cartilage core, and the formation of primary trabeculae.

### 2.2. Vascular Invasion at the Chondro-Osseous Junction

Vascular endothelial cells can invade the epiphyseal cartilage by piercing the incompletely calcified transverse partition of the columns. We demonstrated endomucin-reactive blood vessels invading the chondrocyte lacunae at the chondro-osseous junction [21]. Transmission electron microscopic (TEM) observation verified that the vascular endothelial cells, present in blood vessels close to the cartilaginous matrix, extend their fine cytoplasmic processes into the matrix. The tips of the extended cytoplasmic processes showed fine finger-like structures facing the cartilaginous matrix, suggesting that the apical region of the invading endothelial cells may be partially open. In some observations, cell debris was present inside the blood vessels facing the cartilaginous columns at the chondro-osseous junction, while erythrocytes were found outside the blood vessels. Since the apical region of invading blood vessels might be open, blood vessels could presumably invade the cartilaginous matrix and exclude unnecessary impeditive materials (mainly cellular debris) to avoid accumulation at the junction (Figure 1).

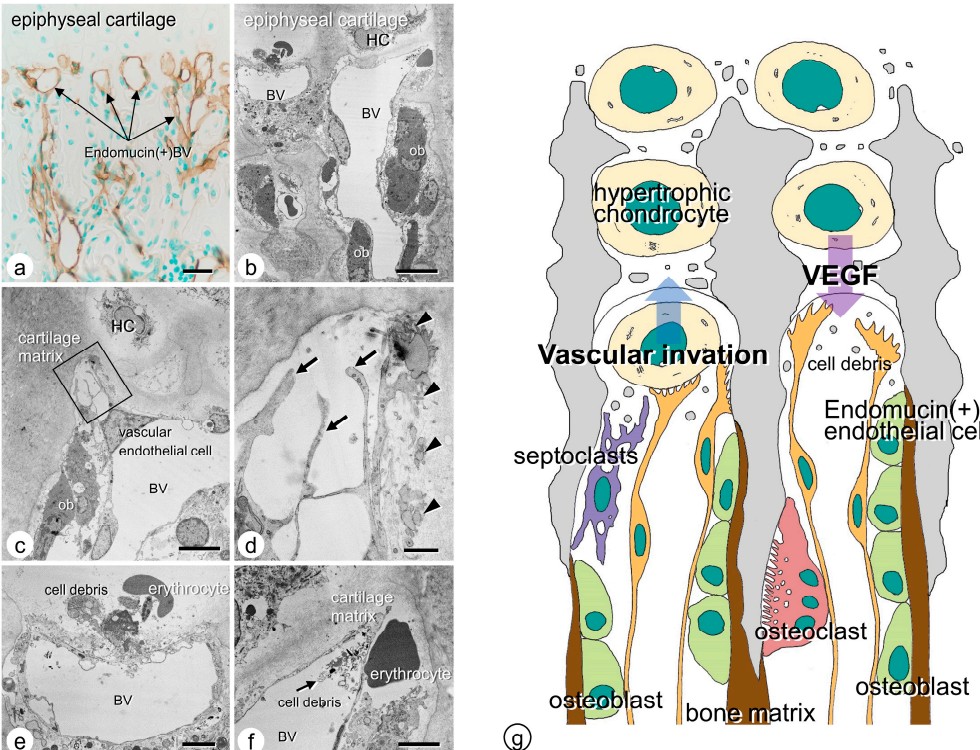

**Figure 1.** Vascular endothelial cells at the chondro-osseous junction. (**a**) Endomucin-immunoreactive (brown color) blood vessels at the chondro-osseous junction under light microscope. (**b**–**f**) TEM images of blood vessels at the chondro-osseous junction. Invading blood vessels are seen beneath the chondrocytic lacunae. (**c**,**d**) When observed under higher magnification as shown in panel c, fine cytoplasmic processes (arrows) are seen extending from the vascular endothelial cell, with invaginations of the cell membranes in the superficial layer of the cartilaginous matrix. (**e**,**f**) Panel e demonstrates cell debris, including erythrocytes from the blood vessels, and panel f reveals an erythrocyte outside the vessel and cell debris in the vessels. (**g**) Schematic design of vascular invasion at the chondro-osseous junction. HP: hypertrophic chondrocyte; BV: blood vessel, ob: osteoblast. Bar, (**a**) 20 μm, (**b**) 10 μm, (**c**,**e**,**f**) 5 μm, (**d**) 1 μm.

*2.3. Osteoclasts' Function at the Chondro-Osseous Junction*

It is well known that osteoclasts, also referred to as chondroclasts, accumulate in the chondro-osseous junction. Osteoclasts at the chondro-osseous junction show intense matrix metalloproteinase (MMP)-9 immunoreactivity [22]. Additionally, MMP-9 immunoreactivity is exhibited in the tips of the vascular endothelial cells facing the cartilaginous matrix, unlike the other areas distant from the chondro-osseous junction [20]. Therefore, osteoclasts and vascular endothelial cells apparently synthesize MMP-9, which dissolves the cartilaginous matrix [23,24]. Vascular invasion rather than osteoclastic resorption seems essential during endochondral ossification. Studies have found that *op/op* mice, *c-fos*$^{-/-}$ mice, and receptor activator of nuclear factor κβ ligand (*Rankl*)$^{-/-}$ mice preserve similar lengths of long bones to those seen in their wild-type counterparts in murine models that lack osteoclasts. However, without osteoclasts, the primary trabeculae form a disorganized but highly connected meshwork in the long bones. As described by Marks and Odgren [25], it seems likely that osteoclastic activity during endochondral ossification resorbs the excess mineralized cartilage matrices and scattered islets of mineralized cartilage in the chondro-osseous junction, enabling the longitudinal arrangement of primary trabeculae. Furthermore, another cell type, septoclasts, also referred to as perivascular cells, may also be involved in vascular invasion during endochondral ossification [26–28]. Septoclasts are positive for Dolichos biflorus agglutinin lectin histochemistry [26] and E-FABP [29,30], featuring well-developed Golgi apparatus and several cytoplasmic lysosomes filled with abundant cathepsin B [27]. We speculate that one major function of septoclasts is to remove excess extracellular organic (non-mineralized) debris that would otherwise interrupt the vascular invasion path into the cartilage, and it is unlikely that osteoclasts are designated to resorb the excess mineralized matrices in the cartilage.

## 3. Ultrastructural Aspects of Matrix Vesicle-Mediated Mineralization in Bone

*3.1. Formation of Crystalline Calcium Phosphates in Matrix Vesicles*

The primary trabeculae resulting from endochondral ossification can be mineralized by osteoblasts. Osteoblasts secrete matrix vesicles enveloped by a plasma membrane (ranging 30–1000 nm in diameter) into the osteoid (incompletely mineralized areas beneath the osteoblasts) [3]. Matrix vesicles are equipped with several enzymes and membrane transporters on the plasma membrane and inside the vesicles, enabling calcium phosphate nucleation and subsequent crystal growth. A crystalline calcium phosphate such as hydroxyapatite crystal [$Ca_{10}(PO_4)_6(OH)_2$] appears inside the matrix vesicles and grows radially, eventually breaking out of the vesicle membrane to form mineralized nodules in a globular assembly of radially oriented hydroxyapatite crystals with a small ribbon-like structure approximately 25 nm wide, 10 nm high, and 50 nm long [31,32].

It seems likely that crystal nucleation begins on the inner leaflet of the vesicle membrane, because the deposition of amorphous mineral crystals is initially observed on the inner leaflet. Acidic phospholipids such as phosphatidylserine and phosphatidylinositol, which have a high affinity for $Ca^{2+}$, are abundantly present in the matrix vesicles and consequently form a stable calcium phosphate–phospholipid complex associated with the inner leaflet of the vesicle membrane [8]. Therefore, it is possible that such complexes may play important roles in crystal nucleation in the matrix vesicles.

*3.2. Mineralized Nodules Develop from Matrix Vesicles*

After crystal formation, matrix vesicles develop mineralized nodules in a globular assembly of needle-like hydroxyapatite crystals (Figure 2). The growth of mineralized nodules appears to be regulated by a large amount of extracellular Ca/Pi and organic materials in the osteoid. To allow the growth of mineralized nodules, many enzymes and transporters on the vesicle membrane may participate in the accumulation of Ca and Pi on the mineralized nodules. However, osteopontin and osteocalcin are suited to the function of regulating mineralization, because they effectively inhibit calcium phosphate nucleation and crystal growth [33,34]. Osteopontin is localized in the periphery of mineralized nodules,

where it might block excessive mineralization [35]. Osteocalcin includes γ-carboxyglutamic acid, which binds to mineral crystals [36–38]. When warfarin, an inhibitor of glutamine residue γ-carboxylation, was administered in our previous study, numerous fragments of needle-shaped mineral crystals were dispersed throughout the osteoid [39] (Figure 3), and γ-carboxylase-deficient mice demonstrated similar abnormality, showing disassembled and scattered crystal minerals in the bones [40]. It seems feasible that γ-carboxylated osteocalcin may bind hydroxyapatite crystals to form and maintain the mineralized nodules.

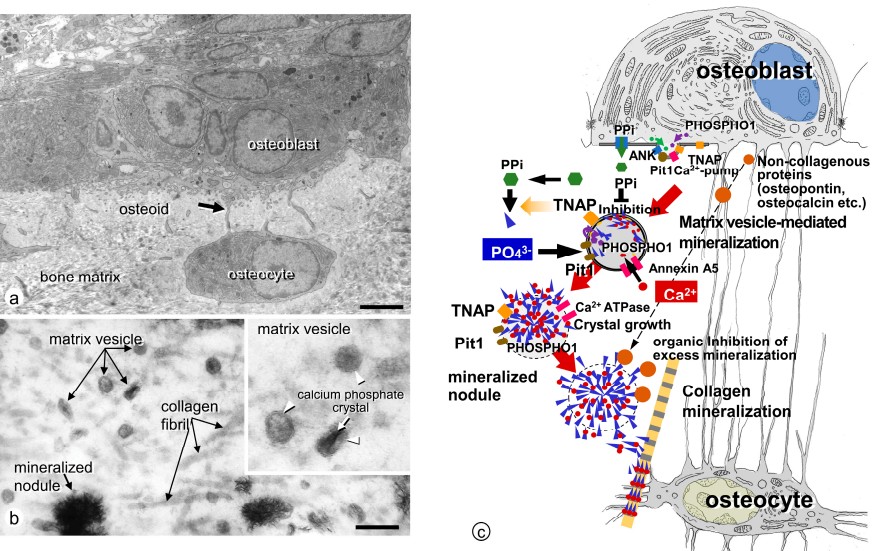

**Figure 2.** Matrix vesicle-mediated bone mineralization by osteoblasts. (**a**,**b**) TEM observation of osteoblasts, osteocytes, and matrix vesicles. (**a**) Mature osteoblasts located on the bone surface (osteoid) connected to osteocytes with their cytoplasmic processes (black arrow). (**b**) At a higher magnification, many matrix vesicles and mineralized nodules are observed. Note the lipid bilayer of the vesicles (white arrowheads) and calcium phosphate crystals (white arrow) in the inset. (**c**) Schematic design of matrix vesicle-mediated bone mineralization. Bar, (**a**) 3 mm, (**b**) 400 nm. Panel C is modified from ref [41].

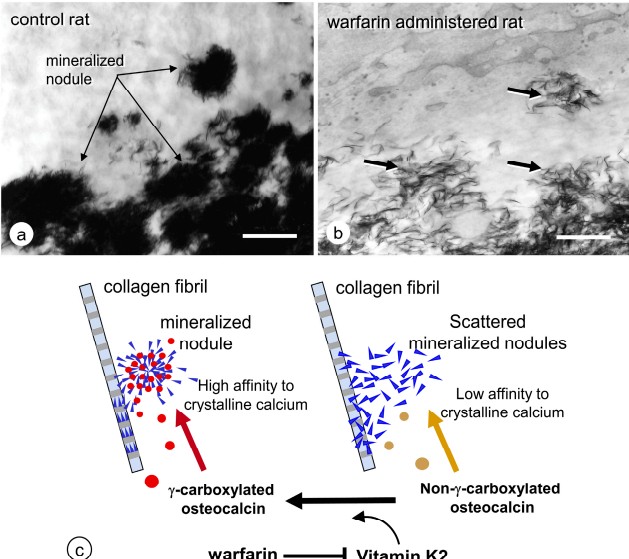

**Figure 3.** Ultrastructure of dispersed mineral crystals in rats administered with warfarin. (**a**) TEM image of mineralized nodules with globular assembled mineral crystals in the control rats. (**b**) The rats administrated with warfarin demonstrate many dispersed mineral crystals (arrows) in the osteoid under TEM. (**c**) Schematic design of forming mineralized nodules by osteocalcin. Bar, 2 mm. Panel C is modified from ref [42].

*3.3. Enzymes and Membrane Transporter Necessary for Matrix Vesicle-Mediated Mineralization in Bone*

Matrix vesicles enable the influx of $Ca^{2+}$ and phosphate ions ($PO_4^{3-}$) by a variety of enzymes and membrane transporters such as tissue nonspecific alkaline phosphatase (TNAP) [6,43–50], ectonucleotide pyrophosphatase/phosphodiesterase 1(ENPP1) [51–53], ankylosis (ANK) [54,55], phosphoethanolamine/phosphocholine phosphatase 1 (PHOS-PHO1) [54–61], and annexins [62]. TNAP, a glycosylphosphatidylinositol-anchored enzyme on the cell membrane, is one of the most important enzymes to initiate mineralization. In bones and cartilages, ENPP1 cleaves the extracellular ATPs into AMPs and pyrophosphate (PPi), and then TNAP hydrolyzes PPi, a phosphorus oxyanion with two phosphorus atoms in a P-O-P linkage, consequently producing $PO_4^{3-}$. The resultant $PO_4^{3-}$ is transported into the matrix vesicles through sodium/phosphate co-transporter type III, also referred to as PiT1. Alternatively, $Ca^{2+}$ can be delivered into the matrix vesicles by passage through annexins. TNAP is expressed not only by mature osteoblasts but also by preosteoblasts (osteoblast precursors), and therefore has been used as an osteoblastic lineage marker.

3.3.1. TNAP

TNAP is localized on the cell membranes of hypertrophic chondrocytes, mature osteoblasts, and preosteoblasts, as well as on the plasma membranes of matrix vesicles [43,44]. However, TNAP is not uniformly localized on the cell membranes of mature bone-synthesizing osteoblasts that possess cell polarity with distinct basolateral and secretory (osteoidal) domains. In one study, although $Ca^{2+}$ transport ATPase was restricted to the osteoidal domain of the osteoblasts, TNAP was predominantly seen on the basolateral domain of the cell membranes [63]. Thus, the membranous domains in bone that feature an abundant TNAP are not matched to the region where TNAP actively serves for matrix vesicle-mediated mineralization. $Tnap^{-/-}$ mice have previously been generated [64,65] to mimic severe hypophosphatasia, with the implication that TNAP is involved in mineralization. $Tnap^{-/-}$ fetuses and neonatal mice have intact bones, but gradually show growth retardation and skeletal deformities. TNAP deficiency not only gives rise to hypomineralization in the skeleton, but also markedly disrupts the alignment of mineral crystals [66]. Thus, TNAP is necessary for normal mineralization and the ultrastructural arrangement of crystalline calcium phosphates in bone. In 2015, the development of the drug asfotase alfa (Strensiq) based on the long-lasting research on TNAP shed a ray of light on the treatment of hypophosphatasia caused by a hereditary mutation of *Tnap* gene [67,68].

3.3.2. ENPP1

ENPP1 cleaves the phosphodiester and pyrophosphate bonds of nucleotides and nucleotide sugars. Analysis of the crystalline structure of ENPP1 showed that nucleotides were accommodated in a pocket formed in the catalytic domain of this molecule, verifying that extracellular ATPs are a substrate for ENPP1 [69]. In bone and cartilage, the catalytic activity of ENPP1 generates PPi, which strongly inhibits mineralization by binding to hydroxyapatite crystals and disrupting their extension [51–53]. However, TNAP cleaves PPi into $PO_4^{3-}$, which is a component of crystalline calcium phosphates in bone. Therefore, balanced interplay between ENPP1 and TNAP seems necessary for bone mineralization [70]. Alternatively, the lack of ENPP1 was proven to be related to the spontaneous mineralization of infantile arteries and periarticular regions [71,72]. In a normal state, therefore, PPi produced by ENPP1 may regulate the growth of hydroxyapatite crystals. In our observations, TNAP was mainly seen in mature osteoblasts and overlying preosteoblasts, while ENPP1 was detected in mature osteoblasts and osteocytes [73]. Genetic ENPP1 dysfunction leading to arterial mineralization may suggest that PPi deficiency or insufficiency can induce osteoblastic differentiation in vascular smooth muscle cells. $Enpp1^{-/-}$ mice, also known as tiptoe walking (*ttw*) mice, undergo ossification of the posterior longitudinal ligament of the spine (OPLL) including progressive ankylosing intervertebral and peripheral joint hyperostosis, as well as articular cartilage mineralization [74–78]. Despite the ectopic min-

eralization, $Enpp1^{-/-}$ mice show reduced serum concentrations of $Ca^{2+}$ and $PO_4{}^{3-}$ as well as significantly elevated serum levels of fibroblast growth factor 23 (FGF23) [78,79]. FGF23 is an osteocyte-derived molecule that inhibits phosphate reabsorption and $1\alpha$-hydroxylase synthesis in the kidney [80–82]. Hence, in $Enpp1^{-/-}$ mice, the induction of Fgf23 mRNA expression, which increases the concentration of serum FGF23, may lead to reductions in the concentrations of $Ca^{2+}$ and $PO_4{}^{3-}$.

### 3.3.3. ANK

ENPP1 can be found not only on the cell surface but also in cytoplasmic regions, generating PPi in both locations. ANK reportedly transports intracellular PPi to the extracellular milieu, i.e., serves as a transmembrane PPi-channeling protein [54,55]. Therefore, it is feasible that ANK-mediated extracellular PPi levels may provide an equivalent balance by disallowing excessive or ectopic mineralization or hypomineralization in various tissues. In previous reports, infants with *Ank* gene mutations exhibited a three to five-fold decrease in extracellular PPi [54], while calcium pyrophosphate (CPP) crystal deposition (CPPD) was elevated in the synovial fluid by gain-of-function mutations in human *ANK* genes [83]. Thus, local PPi production naturally inhibits hydroxyapatite deposition, blocking undesirable mineralization in articular cartilage and other tissues. However, with the loss of ANK activity, extracellular PPi levels attenuate, intracellular PPi levels rise, and unregulated mineralization occurs.

### 3.3.4. PHOSPHO1

PHOSPHO1 is an enzyme abundantly present in bone-forming mature osteoblasts and hypertrophic chondrocytes [56]. Roberts et al. documented that PHOSPHO1 is restricted to the mineralizing regions of the bone and growth plate and plays a role in the initiation of matrix vesicle-mediated mineralization [57]. PHOSPHO1 is reportedly present not only in the cytoplasmic regions of bone-forming osteoblasts and hypertrophic chondrocytes but also in the matrix vesicles. PHOSPHO1 inside the matrix vesicles cleaves $PO_4{}^{3-}$ from phosphatidylcholine and phosphoethanolamine at the inner leaflet of the vesicles' plasma membranes [56–58]. A recent report suggested that phospholipase A2 as well as ENPP6 are also included in matrix vesicles, acting in sequence to produce phosphocholine, which PHOSPHO1 subsequently hydrolyzes into $PO_4{}^{3-}$ [84]. Thus, PHOSPHO1 plays a pivotal role in the increased concentration of $PO_4{}^{3-}$ by cooperating with the $PO_4{}^{3-}$ supply by means of ENPP1/TNAP interplay. Neonatal $Phospho1^{-/-}$ mice demonstrated incomplete mineralization of the bone, often with spontaneous greenstick fractures [59,60]. Millán's team demonstrated that PHOSPHO1 controls TNAP expression in mineralizing cells and is essential for mechanically competent mineralization [59,61]. Taken together, the $PO_4{}^{3-}$ supplementation necessary for matrix vesicle-mediated mineralization appears to be derived at least in part from TNAP/ENPP1 interaction outside the matrix vesicles as well as PHOSPHO1 activity inside the vesicles.

### 3.3.5. Annexins

Annexins are a group of proteins that show high affinity for $Ca^{2+}$ and lipids, serving as ion channels for the transport of $Ca^{2+}$ into the matrix vesicles. Three annexins, annexin A2, A5, and A6, that are abundantly present in vascular endothelial cells, heart, and skeletal muscles, have been discovered in matrix vesicles [62,85–87]. In the initial process of matrix vesicle-mediated mineralization, amorphous calcium phosphates are formed associated with the inner leaflet of the plasma membranes of the matrix vesicles. The annexin A5 might serve as a $Ca^{2+}$ ion channel inside the matrix vesicles. Consequently, transported $Ca^{2+}$ showed strong binding to phosphatidylserine in the inner leaflet of the membrane enclosing the matrix vesicle, which is enriched with anionic lipids [88,89]. Thus, it is feasible that annexin A5 might play an important role in $Ca^{2+}$ transport and subsequent $Ca^{2+}$-dependent phosphatidylserine binding in the matrix vesicles. It is a possibility that the Pi transported through PiT1 present in the membrane could also bind to $Ca^{2+}$ trapped

on the inner leaflet, to form amorphous calcium phosphates. Unexpectedly, *Annexin a5$^{-/-}$* mice did not show skeletal deformity or reduced mineralization, suggesting that other annexins could compensate for the functions of annexin A5. However, further examination is necessary to clarify the precise role of annexins in bone mineralization.

## 4. Regulation of Bone Mineralization by Osteocyte

### 4.1. Erosion of Bone Minerals in the Vicinity of Osteocytes

Osteocytes are located in bone cavities known as osteocytic lacunae, and connect to neighboring osteocytes and osteoblasts on the bone surfaces via fine cytoplasmic processes that run through osteocytic canaliculi [90]. Osteocytes interconnect their cytoplasmic processes via gap junctions, thereby building functional syncytia referred to as the osteocytic lacunar canalicular system (OLCS) [41]. Mature well-mineralized bone develops an OLCS with an orderly arrangement, while immature bone has an irregular and disorganized OLCS [41]. The osteocytic network has been speculated to have roles in multiple processes including mechanical sensing, molecular transport, and regulation of peripheral mineralization [41].

Belanger proposed the concept of osteocytic osteolysis in the 1960s [91], suggesting that osteocytes have the potential not only to erode the peripheral bone minerals but also reversibly to remineralize the bone (Figure 4). This notion was not immediately accepted, however, many researchers have since observed that osteocytes and their canaliculi are involved in the mineral maintenance of the bone matrix [92–99]. The occurrence of osteocytic osteolysis has been reported in cases of PTH administration, including hyperparathyroidism [100,101], during lactation [96,102], in vitamin D receptor deficiency [103], and with sclerostin treatment [104]. During lactation, osteocytes reportedly erode the surrounding bone matrix by exhibiting a pattern of gene expression similar to that of osteoclasts during bone resorption, e.g., an elevation in tartrate-resistant acid phosphatase, cathepsin K, carbonic anhydrase, Na$^+$/H$^+$ exchanger, ATPase H$^+$ transporting lysosomal subunits, and matrix metalloproteinase [96]. Using synchrotron X-ray microscopy, Nango et al. analyzed the degree of bone mineralization in mouse cortical bone around the lacunar canalicular network and reported the dissolution of bone mineral along the osteocyte canaliculi [105]. However, one criticism of the osteocytic osteolysis concept might be that the proteolytic enzymes and acids secreted from the bone-resorbing osteoclasts pass through the osteocytic canaliculi to reach distant osteocytes. Recently, using *Rankl$^{-/-}$* mice, we have obtained microscopic findings that support the idea of osteocytic osteolysis [106]; osteocytic osteolysis is independent of osteoclastic activity and is discernible in mature cortical bone showing a regular distribution of the osteocytic network (Figure 5).

However, several reports have cautioned that (1) large osteocytic lacunae do not always represent the signs of osteocytic osteolysis [107], (2) the vitamin D receptor is not associated with osteocytic osteolysis [108], and (3) despite considerable research, osteocytic osteolysis has continued to be looked upon with skepticism [109]. Nevertheless, many researchers have attempted to elucidate whether osteocytic osteolysis would affect the mechanical properties of bone, and to extend the concept from including merely osteolysis to encompass a remodeling of the osteocytic network's peripheral bone matrix. Recently, Kaya et al. reported that changes in bone mechanical properties induced by lactation and recovery appear to depend predominantly on the volume of osteocytic lacunae and canaliculi, suggesting that tissue-level mechanical properties of cortical bone are rapidly and reversibly modulated by osteocytes in response to physiological challenges [110]. Emami et al. have consistently reported notable canalicular changes following fracture that could affect mechanical properties of bone [111]. Vahidi et al. reported that femoral fracture in mice induced morphological changes of the canalicular network in the contralateral limb, suggesting decreased rates of bone formation and mineralization in the osteocytic lacunar canaliculi. They proposed that changes in canalicular remodeling by osteocytes involve utilization of the mineral from the bones for callus formation and bone repair after a fracture, but this process may also lessen bone quality and systemically elevate the fracture

risk [112]. Osteocytes are the most abundant cells in bone, and the total area of osteocytes and their cytoplasmic processes is much larger than the areas of bone-forming osteoblasts or bone-resorbing osteoclasts. Therefore, osteocytes might be involved in the regulation of mineralization.

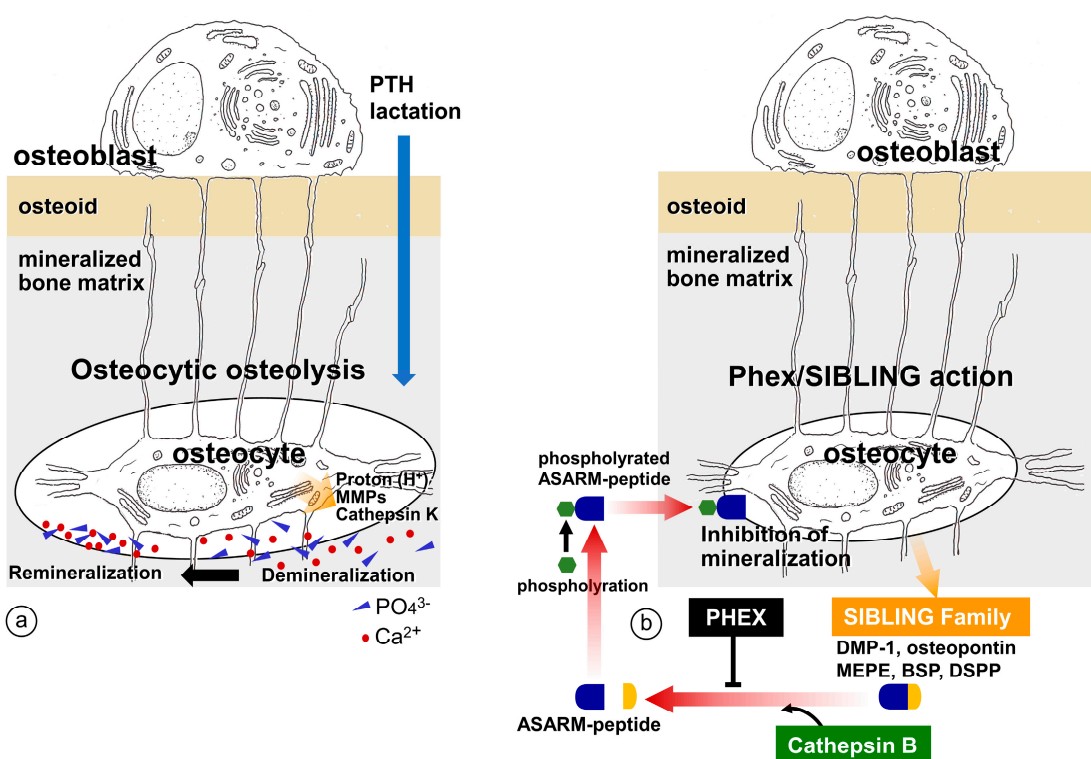

**Figure 4.** Schematic representations of the two hypotheses: (**a**) osteocytic osteolysis and (**b**) regulation of bone mineralization by PHEX/SIBLINGs. During PTH administration or lactation, osteocytes secrete acids and proteolytic enzymes such as cathepsin K and MMPs to erode the surrounding bone. However, osteocytic osteolysis is reversible, so once-eroded bone can be remineralized. In contrast, SIBLINGs such as MEPE, DMP-1, and osteopontin are cleaved by cathepsin B to generate ASARM, which is then phosphorylated to inhibit mineralization. Alternatively, PHEX blocks the inhibition of mineralization.

### 4.2. Regulation of Mineralization by Mediating SIBLING Family

Osteocytes are known to produce many important extracellular molecules, including fibroblast growth factor 23 (FGF23), small integrin-binding ligand N-linked glycoprotein (SIBLING) family proteins, and phosphate-regulating gene with homologies to endopeptidases on the X chromosome (PHEX). Through these molecules, osteocytes can regulate bone mineralization in two different manners: (1) systemic regulation of serum Pi by FGF23 in the kidney; and (2) local regulation of mineral crystal growth by PHEX/SIBLING family.

For systemic regulation of serum Pi, FGF23 secreted from osteocytes is circulated to reach the kidneys, where it binds to the receptor complex of fibroblast growth factor receptor Ic (FGFR1c) and αklotho expressed in the proximal renal tubules, to inhibit sodium/phosphate co-transporter type IIa/IIc (NaPi IIa/IIc). Since NaPi IIa/IIc reabsorb phosphate ions in the proximal renal tubules, FGF23 reduces the serum Pi concentration [80–82]. Human X-linked hypophosphatemia (XLH), one of the FGF23-related causes of hypophosphatemic rickets or osteomalacia in children and osteomalacia in adults, is caused by loss-of-function mutations in PHEX resulting in the elevated circulation of FGF23 and markedly decreased bone mineralization. This may indicate that the osteocyte-derived hormone FGF23, along with its function in the kidneys, may play a pivotal role in the systemic regulation of bone mineralization.

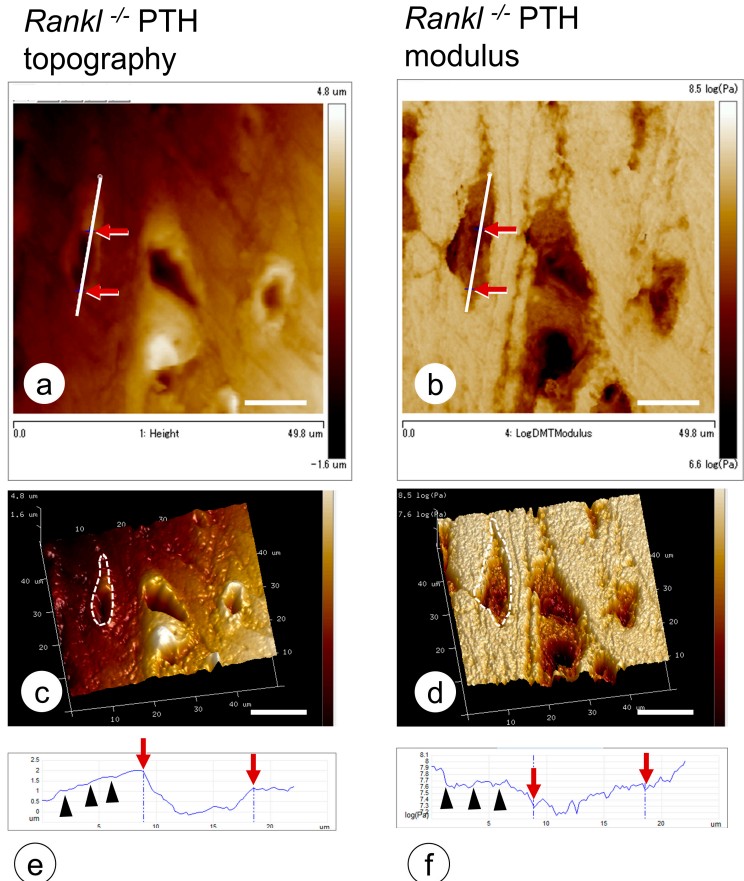

**Figure 5.** Nano-indentation by atomic force microscopy on bone matrix surrounding osteocytic lacunae. (**a**,**c**) Topography of osteocytic lacunae in the femoral cortical bone of the PTH-administered *Rankl*$^{-/-}$ mice. (**b**,**d**) Elastic modulus of the osteocytic lacunae in the femoral cortical bone of the PTH-administered *Rankl*$^{-/-}$ mice. Red arrows along the white lines in a and b are matched with the red arrows in the graphs (**e**,**f**). Note the slightly expanded diameters in the three-dimensional images of the elastic modulus (compare the dotted circles in (**c**,**d**)), and that the index of the elastic modulus is lower than the topography in the graph (black arrowheads). Bar, 20 mm.

In contrast to systemic regulation of serum Pi and bone mineralization, osteocytes appear to regulate mineralization in the periphery of the osteocytic lacunae. Dentin matrix protein-1 (DMP-1), which is secreted by osteocytes, has high potential to bind Ca$^{2+}$ and is postulated to play a role in the mineralization of the peripheral bone matrix of osteocytes [113]. The SIBLING family includes DMP-1, matrix extracellular phosphoglycoprotein (MEPE), osteopontin, bone sialoprotein (BSP), and dentin sialophosphoprotein (DSPP), which are encoded by a gene located on human chromosome 4q21 and mouse chromosome 5q21 [114,115]. We considered the possibility that the interaction between PHEX and the SIBLING family might regulate mineralization in the periphery of the osteocytic lacunae (Figure 4). For instance, MEPE secreted by osteocytes is cleaved by cathepsin B to release the carboxy terminal region, a novel functional domain referred to as the acidic serine-rich and aspirate-rich motif (ASARM) [116,117]. The resultant ASARM peptides are then phosphorylated to inhibit bone mineralization [117]. However, MEPE also binds to PHEX, forming the MEPE-PHEX complex. In this situation, cathepsin B is unable to cleave the MEPE-PHEX complex, which therefore blocks the synthesis of ASARM, so no phosphorylated ASARM inhibits mineralization, and normal mineralization is thereby attained [118]. It has been reported that the phosphorylated ASARM peptide of osteopontin inhibits mineralization in a phosphorylation-dependent manner, and PHEX disturbs the inhibition of mineralization [119]. These findings implicate the possibility that osteocyte-derived SIBLINGs may regulate peripheral bone mineralization by cooperating with PHEX.

This idea is supported by the observation that the absence of DMP-1 results in rickets or osteomalacia in mice [120] and autosomal recessive hypophosphatemic rickets or osteomalacia (ARHR) in human patients [121]. However, PHEX/SIBLINGs are usually associated with congenital deformities, rickets, and osteomalacia, and therefore it is necessary to elucidate whether PHEX/SIBLINGs play an important role in the physiological regulation of bone mineralization in a healthy state.

## 5. Conclusions

During endochondral ossification, hypertrophic chondrocytes secrete matrix vesicles into the intercolumnar septa but not the transverse partitions of the cartilage columns; this allows vascular invasion into the epiphyseal cartilage and subsequent osteoblastic bone formation in the mineralized cartilage core. Thus, endochondral ossification is finely tuned by the cellular interplay at the chondro-osseous junction. To achieve matrix vesicle-mediated mineralization, many enzymes and membrane transporters including TNAP, ENPP1, PiT1, PHOSPHO1, annexins, and others are involved in the influx of $Ca^{2+}$/Pi and the regulation of calcium phosphate crystal growth. In addition to their role in osteoblastic primary mineralization, osteocytes have recently been shown to regulate bone mineralization, presumably by controlling the synthesis of PHEX/SIBLING, as well as osteocytic osteolysis. Thus, normal mineralization is maintained by the orchestrated activities of bone cells.

**Author Contributions:** Conceptualization, writing—original draft preparation, T.H.; investigation; T.H., H.H., T.H., T.Y., T.M. and Y.M.; writing—review and editing, N.A.; supervision and funding acquisition, T.H. and N.A. All authors have read and agreed to the published version of the manuscript.

**Funding:** This study was partially supported by grants from the Japanese Society for the Promotion of Science (JSPS, 22K09911 to T.H., 21K16928 to H.H., and 21H03103 to N.A.) and a grant-in-aid for young scientists provided by the Uehara Memorial Foundation (202110102 to T.H.).

**Institutional Review Board Statement:** Not applicable.

**Informed Consent Statement:** Not applicable.

**Data Availability Statement:** Not applicable.

**Conflicts of Interest:** The authors declare no conflict of interest.

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
