# Peer review of "Histological Assessment of Endochondral Ossification and Bone Mineralization"

_endocrines, doi:10.3390/endocrines4010006_

Round 1

Reviewer 1 Report

While this paper does not introduce much novel data to the understanding of the physiology of the epiphyseal growth plate, it is one of the most well-written and comprehensive summaries of this amazing physiological "snapshot" of a localized developmental program. The authors very skillfully integrates the literature in the narrative to lend credence to their findings. This should become a staple in the bone literature or in training our students.

That said, it is incredibly confusing as to whether there is any original data in this manuscript. There is no methods section and if the data presented is a duplication of work from other labs- there is no citation in the figure legends. This needs to be clarified and the paper presented as a review article and not an original publications.

The paper does not address many of the regulatory factors that drive long bone growth (FGFs and FGFR3, IGF-1, PTHrp, IHH-signaling) but may be considered as a follow up to this study, especially with the advent of single cell analysis of growth plate chondrocytes.

Minor:

Line 396: Human X-linked hypophosphatemia (XLH), one of the FGF23-related causes of hypophosphatemic rickets/osteomalacia in children and osteomalacia in adults

Figure 4 text needs a space inserted.

Author Response

Our responses to the reviewer #1

Reviewer’s general comments
While this paper does not introduce much novel data to the understanding of the physiology of the epiphyseal growth plate, it is one of the most well-written and comprehensive summaries of this amazing physiological "snapshot" of a localized developmental program. The authors very skillfully integrates the literature in the narrative to lend credence to their findings. This should become a staple in the bone literature or in training our students.
    That said, it is incredibly confusing as to whether there is any original data in this manuscript. There is no methods section and if the data presented is a duplication of work from other labs- there is no citation in the figure legends. This needs to be clarified and the paper presented as a review article and not an original publications. The paper does not address many of the regulatory factors that drive long bone growth (FGFs and FGFR3, IGF-1, PTHrp, IHH-signaling) but may be considered as a follow up to this study, especially with the advent of single cell analysis of growth plate chondrocytes.

Our response
First, we would like to express our appreciation to the reviewer for providing invaluable suggestions for our review article. We agree with the reviewer. This review article has been written based on not only the investigations by other researchers but also our reports. However, as the reviewer suggested, it is important to clarify which parts are our new insights and to properly express the original content in this review article. Now, we have attempted to clearly present our recent reports and our new insights in the manuscript, and also provide citations in the figure legends as suggested. Regarding the regulatory factors for the long bone growth (FGFR3, PTHrP, IHH-signaling during cartilage growth), we would like to add some sentences to the paragraph “2.1. Cartilage mineralization by hypertrophic chondrocytes” and the references as follows.

Line 73-89
Chondrocytes form the longitudinal columns in the proliferative and hypertrophic zone, but the proliferative chondrocytes synchronously enlarge for entering into hypertrophic phenotype [1]. Parathyroid hormone (PTH)-related peptide (PTHrP) has been reported to regulate hypertrophic differentiation of chondrocytes by mediating the Indian Hedgehog (IHH)/PTHrP negative feedback [9]. IHH expressed in the prehypertrophic zone (the upper region of hypertrophic zone) stimulates PTHrP expression in the early differentiation stage of chondrocytes. PTHrP promotes the proliferation activity of chondrocytes by binding to the common receptor of PTH and PTHrP (PTH/PTHrP receptor) in the proliferative zone. PTHrP alternatively inhibits the hypertrophic phenotype of chondrocytes, and then, IHH expression is turned off. In addition to IHH/PTHrP negative feedback, another important regulatory factor in chondrocyte proliferation is fibroblast growth factor receptor 3 (FGFR3). Point mutations in FGFR3 cause achondroplasia and thanatophoric dysplasia by continuous activation of the transcription factor Stat1 [10, 11]. FGFR3 signaling has also been proposed to increase the pool of proliferating cells by stimulating chondrocytes in the resting zone and promoting their transit to the proliferative zone [12, 13]. Thus, the action of IHH/PTHrP and FGFR3 may be essential for chondrocyte proliferation and differentiation [14]

References
9.    Ohba, S. Hedgehog Signaling in Skeletal Development: Roles of Indian Hedgehog and the Mode of Its Action. Int J Mol Sci. 2020, 21, 6665. 
10.    Tavormina, PL.; Shiang, R.; Thompson, LM.; Zhu, YZ.; Wilkin, DJ.; Lachman, RS.; Wilcox, WR.; Rimoin, DL.; Cohn, DH.; Wasmuth, JJ. Thanatophoric dysplasia (types I and II) caused by distinct mutations in fibroblast growth factor receptor 3. Nat Genet. 1995, 9, 321-328. 
11.    Su, WC.; Kitagawa, M.; Xue, N.; Xie, B.; Garofalo, S.; Cho, J.; Deng, C.; Horton, WA.; Fu, XY. Activation of Stat1 by mutant fibroblast growth-factor receptor in thanatophoric dysplasia type II dwarfism. Nature. 1997, 386, 288-292. 
12.    Peters, K.; Ornitz, D.; Werner, S.; Williams, L. Unique expression pattern of the FGF receptor 3 gene during mouse organogenesis. Dev. Biol. 1993, 155, 423-430.
13.    Deng, C.; Wynshaw-Boris, A.; Zhou, F.; Kuo, A.; Leder, P.  Fibroblast growth factor receptor 3 is a negative regulator of bone growth. Cell. 1996, 84, 911-921.
14.    Amizuka, N.; Davidson, D.; Liu, H.; Valverde-Franco, G.; Chai, S.; Maeda, T.; Ozawa, H.; Hammond, V.; Ornitz, DM.; Goltzman, D.; Henderson, JE. Signalling by fibroblast growth factor receptor 3 and parathyroid hormone-related peptide coordinate cartilage and bone development. Bone. 2004, 34, 13-25. 

Reviewer’s specific comment #1
Line 396: Human X-linked hypophosphatemia (XLH), one of the FGF23-related causes of hypophosphatemic rickets/osteomalacia in children and osteomalacia in adults

Our responses
Thank you for your critical comments. We added “Human X-linked hypophosphatemia (XLH), one of the FGF23-related causes of hypophosphatemic rickets/osteomalacia in children and osteomalacia in adults” in the corresponding sentence (Line529-530). In addition, the paragraph“4. Regulation of bone mineralization by osteocyte”has been completely revised and improved. 

Reviewer’s specific comment #2
Figure 4 text needs a space inserted.

Our responses
Thank you for your suggestion. We inserted a space ahead of the text in Figure 4. In addition, we added new Figures 4 and 5 in the revised manuscript. Please note that the previous Figure 4 has changed to Figure 5.

We would like to express our sincere appreciation to reviewer #1. We believe that your invaluable suggestions have helped us significantly improve our paper. 

Reviewer 2 Report

The manuscript endocrine-2017286 by Hasegawa T et al attempts to provide an updated overview of the endochondral ossification process and bone mineralization.

The manuscript encompasses relevant information from historical and updated literature, but it is quite difficult to follow and need to be reorganized in its structure. A review article should clearly guide the reader through the available literature to understand the main addressed topic, unfortunately here the authors are presenting several information without critical description of the main message. Schematic figures need to be provided together with the histological figures which indeed are not really appropriated to follow the text description.

Comments

Line 293: please fix-plasma not plasm

Line 304: please fix- the skeleton not theskeleton

Line 329-333; it is unclear the link among Enppi, Calcium, Pi and FFG23

Line 340: please fix- infants not Infants

Line 360-362: the sentence is difficult to understand, please rephrase

Line 365-367: the description of the matrix vesicle enzymes should be provided before

Line 367: it is unclear the meaning of “reverent” here

Line 378: please substitute equipped with present

Paragraph 4.1 need to be fully revised since the role and function of PHEX and SIBLING is very difficult to follow.

Author Response

Our responses to the reviewer #2

Reviewer’s general comments
The manuscript endocrine-2017286 by Hasegawa T et al attempts to provide an updated overview of the endochondral ossification process and bone mineralization. The manuscript encompasses relevant information from historical and updated literature, but it is quite difficult to follow and need to be reorganized in its structure. A review article should clearly guide the reader through the available literature to understand the main addressed topic, unfortunately here the authors are presenting several information without critical description of the main message. Schematic figures need to be provided together with the histological figures which indeed are not really appropriated to follow the text description.

Our response
First of all, we would like to express our appreciation to the reviewer for providing invaluable suggestions. We agree that the text was difficult to follow. To address this, as suggested by reviewer, we have added schematic figures, which could be helpful for understanding the histological images in Figures 1–4. We also provided additional explanation in the figure legends. We believe that, because of your invaluable suggestions, our paper has been significantly improved. We thank you very much for your critical comments.

Reviewer’s specific comment #1
Line 293: please fix- plasma not plasm
Line 304: please fix- the skeleton not theskeleton
Line 340: please fix- infants not Infants

Our responses
Thank you for pointing out these mistakes. We have now corrected them, as suggested, in the revised manuscript.

Reviewer’s specific comment #2
Line 329-333; it is unclear the link among Enppi, Calcium, Pi and FFG23

Our responses
Again, thank you for your critical comment. As suggested, we have added new content on the relationship between ENPP1, calcium, phosphate, and FGF23 in the revised manuscript, as quoted below.

Line 332-337
Despite the ectopic mineralization, Enpp1−/− mice show a reduced serum concentration of Ca2+ and PO43− as well as significantly elevated serum levels of fibroblast growth factor 23 (FGF23) [78, 79]. FGF23 is an osteocyte-derived molecule, which inhibits phosphate reabsorption and 1α-hydroxylase synthesis in the kidney [80-82]. Hence, in Enpp1−/− mice, the induction of Fgf23 mRNA expression, which increases the concentration of serum FGF23, may lead to a reduction in the concentrations of Ca2+ and PO43−

Reviewer’s specific comment #3
Line 360-362: the sentence is difficult to understand, please rephrase

Our responses
Thank you for your suggestion. We have rephrased the sentences as follows.

Line 367-369
Taken together, PO43− supplementation needed for matrix vesicle-mediated mineralization appears to be derived from, at least in part, TNAP/ENPP1 interaction outside the matrix vesicles as well as PHOSPHO1 activity inside the vesicles.

Reviewer’s specific comment #4
Line 365-367: the description of the matrix vesicle enzymes should be provided before

Our responses
Thank you for your advice. We have briefly described the matrix vesicle enzymes and membrane transporters in the first paragraph of section 3.3. As suggested by the reviewer, we have removed the sentences “Matrix vesicles include a variety of enzymes and proteins such as PHOSPHO1 [62-67], TNAP [6, 36, 68-73], ….Among these molecules,”. Regarding MMP-3, carbonic anhydrase II, phospholipase A2, and lactate dehydrogenase, we would like to remove MMP-3, carbonic anhydrase II, and lactate dehydrogenase from the revised manuscript (Phospholipase A2 is described in the section of PHOSPHO1), because these enzymes are less investigated compared to TNAP, ENPP1 and PHOSPHO1 in the context of matrix vesicle-mediated mineralization. Thus, we revised the sentences as follows.

Line 277-291 
3.3. Enzymes and membrane transporter necessary for matrix vesicle-mediated mineralization in bone
Matrix vesicles enable the influx of Ca2+ and phosphate ions (PO43−) by a variety of enzymes and membrane transporters such as tissue nonspecific alkaline phosphatase (TNAP) [6, 43-50], ectonucleotide pyrophosphatase/phosphodiesterase 1(ENPP1) [51-53], ankylosis (ANK) [54, 55], phosphoethanolamine/phosphocholine phosphatase 1 (PHOSPHO1) [54-61], annexins [62]. TNAP, a glycosylphosphatidylinositol-anchored enzyme on the cell membrane, is one of the most important enzymes to initiate mineralization. In bones and cartilages, ENPP1 cleaves the extracellular ATPs into AMPs and pyrophosphate (PPi), and then, TNAP hydrolyzes PPi, a phosphorus oxyanion with two phosphorus atoms in a P-O-P linkage, consequently producing PO43−. The resultant PO43− is transported into the matrix vesicles through sodium/phosphate co-transporter type III, also referred to as PiT1. Alternatively, Ca2+  can be delivered inside the matrix vesicles by passing through an-nexins. TNAP is not only expressed by mature osteoblasts but also by preosteoblasts (osteoblast precursors), and therefore, has been used as an osteoblastic lineage marker.

Line 367-371 
3.3.5. Annexins
Annexins are a group of proteins that show high affinity for Ca2+ and lipids, serving as ion channels for the transport of Ca2+ into the matrix vesicles. Three annexins, annexin A2, A5, and A6, which are abundantly present in vascular endothelial cells, heart, and skeletal muscles, have been observed in matrix vesicles [62, 85-87].

Reviewer’s specific comment #5
Line 367: it is unclear the meaning of “reverent” here

Our responses
Thank you for your comment. We removed “reverent” from this sentence. 

Reviewer’s specific comment #6
Line 378: please substitute equipped with present

Our responses
Thank you for your advice. We substituted “equipped” with “present”.

Reviewer’s specific comment #7
Paragraph 4.1 need to be fully revised since the role and function of PHEX and SIBLING is very difficult to follow.

Our responses
We that the reviewer for this critical comment, which we totally agree with. Therefore, we have rewritten the paragraph “4. Regulation of bone mineralization by osteocyte” and also changed the order of “4.2. Erosion of bone minerals in the vicinity of osteocytes” and “4.1 Osteocytic re-mineralization by mediating SIBLING family”. With this reordering of the paragraphs, we believe it is now easier for readers to follow the content. In addition, we added a schematic figure to help the readers visualize the osteocytic regulation of bone mineralization. Please find the revised version of the manuscript including a new Figure 4.

Line 385-560
4.1. Erosion of bone minerals in the vicinity of osteocytes
Osteocytes are located in bone cavities named osteocytic lacunae and connect to neighboring osteocytes and osteoblasts on the bone surfaces via fine cytoplasmic processes that run through osteocytic canaliculi [90]. Osteocytes interconnect their cytoplasmic processes via gap junctions, thereby building functional syncytia referred to as osteocytic lacunar-canalicular system (OLCS) [41]. Mature, well-mineralized bone develops an orderly arranged OLCS, while immature bone features irregular, disorganized OLCS [41]. The osteocytic network has been speculated to have roles in multiple processes including mechanical sensing, molecular transport, and regulation of peripheral mineralization [41]. 
Belanger proposed the concept of osteocytic osteolysis in the 1960s [91], where osteocytes are suggested to have the potential to not only erode the peripheral bone minerals but also reversibly re-mineralize the bone (Figure 4). This concept was not immediately accepted; however, since then, many researchers have observed that osteocytes and their canaliculi are involved in the mineral maintenance of the bone matrix [92-99]. The occurrence of osteocytic osteolysis has been reported in cases of PTH administration, including hyperparathyroidism [100, 101], during lactation [96, 102], in vitamin D receptor deficiency [103], and with sclerostin treatment [104]. During lactation, osteocytes reportedly erode the surrounding bone matrix by exhibiting a gene expression pattern similar to that of osteoclasts during bone resorption, e.g., an elevation in tartrate‐resistant acid phosphatase, cathepsin K, carbonic anhydrase, Na+/H+ exchanger, ATPase H+ transporting lysosomal subunits, and matrix metalloproteinase [96]. Using synchrotron X-ray microscopy, Nango et al. analyzed the degree of bone mineralization in mouse cortical bone around the lacunar-canalicular network and reported the dissolution of bone mineral along the osteocyte canaliculi [105]. However, one criticism on the osteocytic osteolysis concept might be that the proteolytic enzymes and acids secreted from the bone-resorbing osteoclasts pass through the osteocytic canaliculi to reach distant osteocytes. Recently, using Rankl-/- mice, we have obtained several microscopic findings that strengthen the osteocytic osteolysis concept [106]; osteocytic osteolysis is independent of osteoclastic activity and is discernible in mature cortical bone showing a regular distribution of the osteocytic network (Figure 5). 
However, several reports cautioned that (1) large osteocytic lacunae do not always represent the signs of osteocytic osteolysis [107], (2) the vitamin D receptor is not associated with osteocytic osteolysis [108], and (3) despite much research, osteocytic osteolysis has continued to be looked upon with skepticism [109]. Nevertheless, many researchers have attempted to elucidate whether osteocytic osteolysis would affect bone mechanical properties and to extend the concept from merely osteolysis to remodeling of the osteocytic network’s peripheral bone matrix. Recently, Kaya et al., reported that changes in the bone mechanical properties induced by lactation and recovery appear to depend predominantly on the volume of osteocytic lacunae and canaliculi, suggesting that tissue-level cortical bone mechanical properties are rapidly and reversibly modulated by osteocytes in response to physiological challenges [110]. Emami et al. have consistently reported notable canalicular changes following fracture that could affect bone mechanical properties [111]. Vahidi et al. have found that femoral fracture in mice induce morphological changes of the canalicular network in the contralateral limb, suggesting decreased rates of bone formation and mineralization in the osteocytic lacunar canaliculi. They proposed that changes in canalicular remodeling by osteocytes is a process by which the mineral is utilized from the bones for callus formation and bone repair after a fracture, but this process may also lessen the bone quality and elevate the fracture risk systemically [112]. Osteocytes are the most abundant cells in bone, and the total area of osteocytes and their cytoplasmic processes is much larger than the areas of bone-forming osteoblasts or bone-resorbing osteoclasts. Therefore, osteocytes might be involved in the regulation of mineralization.
4.2. Regulation of mineralization by mediating SIBLING family
Osteocytes are known to produce many important extracellular molecules, including fibroblast growth factor 23 (FGF23), small integrin-binding ligand N-linked glycoprotein (SIBLING) family proteins, and phosphate-regulating gene with homologies to endopeptidases on the X chromosome (PHEX). Through these molecules, osteocytes can regulate bone mineralization in two different manners: (1) systemic regulation of serum Pi by FGF23 in the kidney and (2) local regulation of mineral crystal growth by PHEX/SIBLING family.
For systemic regulation of serum Pi, FGF23 secreted from osteocytes is circulated to reach the kidneys, where it binds to the receptor complex of fibroblast growth factor receptor Ic (FGFR1c) and αklotho expressed in the proximal renal tubules to inhibit sodium/phosphate co-transporter type IIa/IIc (NaPi IIa/IIc). Since NaPi IIa/IIc reabsorb phosphate ions in the proximal renal tubules, FGF23 reduces the serum Pi concentration [80-82]. Human X-linked hypophosphatemia (XLH), one of the FGF23-related caused of hypophosphatemic rickets/osteomalacia in children and osteomalacia in adults, is caused by the loss-of-function mutations in PHEX, which result in the elevated circulation of FGF23 and markedly decreased bone mineralization. This may indicate that the osteocyte-derived hormone FGF23, along with its role in the kidneys, may play a pivotal role in the systemic regulation of bone mineralization.
In contrast to systemic regulation of serum Pi and bone mineralization, osteocytes appear to regulate mineralization in the periphery of the osteocytic lacunae. Dentin matrix protein 1 (DMP1), which is secreted by osteocytes, has a high potential to bind Ca2+ and is postulated to play a role in the mineralization of the peripheral bone matrix of osteocytes [113]. SIBLING family includes DMP1, matrix extracellular phosphoglycoprotein (MEPE), osteopontin, bone sialoprotein (BSP), and dentin sialophosphoprotein (DSPP), which are encoded by a gene located on human chromosome 4q21 and mouse chromosome 5q21 [114, 115]. We considered the possibility that the interaction between PHEX and SIBLING family might regulate mineralization in the periphery of osteocytic lacunae (Figure 4). For instance, MEPE secreted by the osteocytes is cleaved by cathepsin B to release the carboxy-terminal region, which is a novel functional domain referred to as the acidic serine-rich and aspirate-rich motif (ASARM) [116, 117]. The resultant ASARM peptides are then phosphorylated to inhibit bone mineralization [117]. However, MEPE also binds to PHEX, forming the MEPE-PHEX complex. If so, cathepsin B can hardly cleave the MEPE-PHEX complex, which therefore blocks the synthesis of ASARM, and no phosphorylated ASARM would inhibit mineralization, thereby allowing normal mineralization [118]. It has been reported that, the phosphorylated ASARM peptide of osteopontin inhibits mineralization in a phosphorylation-dependent manner, and PHEX disturbs the inhibition of mineralization [119]. These findings implicate the possibility that osteocyte-derived SIBLINGs would regulate peripheral bone mineralization by cooperating with PHEX. This idea may be supported by the observation that the absence of DMP1 results in rickets or osteomalacia in mice [120] and autosomal recessive hypophosphatemic rickets/osteomalacia (ARHR) in human patients [121]. However, PHEX/SIBLINGs are usually associated with congenital deformities rickets and osteomalacia, and therefore, it is necessary to elucidate whether PHEX/SIBLINGs play an important role in the physiological regulation of bone mineralization in a healthy state.

Figure 4 legend
Figure 4 Schematic representations of the two hypothesis (a) osteocytic osteolysis and (b) regulation of bone mineralization by PHEX/SIBLINGs. During PTH administration or lactation, osteocytes secret acids and proteolytic enzymes such as cathepsin K and MMPs to erode the surrounding bone. However, osteocytic osteolysis is reversible, so that, once-eroded bone can be re-mineralized. In contrast, SIBLINGs such as MEPE, DMP-1, and osteopontin are cleaved by cathepsin B to generate ASARM, which is then phosphorylated to inhibit mineralization. Alternatively, PHEX blocks the inhibition of mineralization. 

Finally, we would like to express our sincere appreciation to reviewer #2. We believe that your invaluable suggestions have helped us significantly improve our manuscript. 

Round 2

Reviewer 2 Report

The authors appropriately adreessed all the comments.